# Reinforcement Learning Guided Semi-Supervised Learning

**Marzi Heidari[1], Hanping Zhang[1], Yuhong Guo[1,2]**
[1]School of Computer Science, Carleton University, Ottawa, Canada
[2]CIFAR AI Chair, Amii, Canada
{marziheidari@cmail, jagzhang@cmail, yuhong.guo}.carleton.ca

## Abstract

In recent years, semi-supervised learning (SSL) has gained significant attention due to its ability to leverage both labeled and unlabeled data to improve model performance, especially when labeled data is scarce. However, most current SSL methods rely on heuristics or predefined rules for generating pseudo-labels and leveraging unlabeled data. They are limited to exploiting loss functions and regularization methods within the standard norm. In this paper, we propose a novel Reinforcement Learning (RL) Guided SSL method, RLGSSL, that formulates SSL as a one-armed bandit problem and deploys an innovative RL loss based on weighted reward to adaptively guide the learning process of the prediction model. RLGSSL incorporates a carefully designed reward function that balances the use of labeled and unlabeled data to enhance generalization performance. A semi-supervised teacher-student framework is further deployed to increase the learning stability. We demonstrate the effectiveness of RLGSSL through extensive experiments on several benchmark datasets and show that our approach achieves consistent superior performance compared to state-of-the-art SSL methods.

## 1  Introduction

Semi-supervised learning (SSL) is a significant research area in the field of machine learning, addressing the challenge of effectively utilizing limited labeled data alongside abundant unlabeled data. SSL techniques bridge the gap between supervised and unsupervised learning, offering a practical solution when labeling large amounts of data is prohibitively expensive or time-consuming. The primary goal of SSL is to leverage the structure and patterns present within the unlabeled data to improve the learning process, generalization capabilities, and overall performance of the prediction model. Over the past few years, there has been considerable interest in developing various SSL methods, and these approaches have found success in a wide range of applications, from computer vision [1] to natural language processing [2] and beyond [3, 4].

Within the SSL domain, a range of strategies has been devised to effectively utilize the information available in both labeled and unlabeled data. Broadly, SSL approaches can be categorized into three key paradigms: regularization-based, mean-teacher-based, and pseudo-labeling methodologies. Regularization-based approaches form a fundamental pillar of SSL [5–7]. These methods revolve around the core idea of promoting model robustness against minor perturbations in the input data. A quintessential example in this category is Virtual Adversarial Training (VAT) [5]. VAT capitalizes on the introduction of adversarial perturbations to the input space, thereby ensuring the model's predictions maintain consistency. The second category, Mean-teacher based methods, encapsulates a distinct class of SSL strategies that leverage the concept of temporal ensembling. This technique aids in the stabilization of the learning process by maintaining an exponential moving average of model parameters over training iterations. Mean Teacher [8] notably pioneered this paradigm with a

Mean Teacher model, illustrating its efficacy across numerous benchmark tasks. Lastly, the category of pseudo-labeling approaches has attracted attention due to its simplicity and effectiveness. These methods employ the model's own predictions on unlabeled data as "pseudo-labels" to augment the training process. The MixMatch [1] framework stands as one of the leading representatives of this category, demonstrating the potential of these methods in the low-data regime.

Despite these advancements, achieving high performance with limited labeled data continues to be a significant challenge in SSL, often requiring intricate design decisions and careful coordination of multiple loss functions. Recently RL has been increasingly used in fine-tuning complex models with non-differentiable reward functions. This application establishes a reinforced alignment with the learning objective and enhances generalization to standard supervised learning scenarios [9–12]. Enthusiastic about the potential of enhanced generalizability introduced by a non-differentiable RL reward to SSL, we propose to approach SSL outside the conventional design norms by developing a Reinforcement Learning Guided Semi-Supervised Learning (RLGSSL) method, which brings a fresh perspective to SSL. We employ Reinforcement Learning (RL) to optimize the generation of pseudo-labels in Semi-Supervised Learning (SSL). Conventional methods in SSL often face problems such as overfitting and difficulties in creating accurate pseudo-labels. RL introduces advantages in exploration capabilities and adeptly manages non-differentiable operations by considering the predictor as a policy function.

In RLGSSL, we formulate SSL as a bandit problem, a special case of reinforcement learning, where the prediction model serves as the policy function, and soft pseudo-labeling acts as the actions. We define a simple reward function that balances the use of labeled and unlabeled data and improves generalization capacity by leveraging linear data interpolation, while the prediction model is trained under a policy gradient framework to maximize the policy-output weighted reward. Formulating the SSL problem as such an RL task allows our approach to dynamically adapt and respond to the data. Moreover, we further deploy a teacher-student learning framework to enhance the stability of learning. Additionally, we integrate a supervised learning loss to improve and accelerate the learning process. This new SSL framework has the potential to pave the way for more robust, flexible, and adaptive SSL methods. We evaluate the proposed method through extensive experiments on benchmark datasets. The contribution of this work can be summarized as follows:

- We propose RLGSSL, a novel Reinforcement Learning based approach that effectively tackles SSL by leveraging RL's power to learn effective strategies for generating pseudo-labels and guiding the learning process.
- We design a prediction assessment reward function that encourages the learning of accurate and reliable pseudo-labels while maintaining a balance between the usage of labeled and unlabeled data.
- We develop an innovative RL loss that allows reward from pseudo-labels to be incorporated into SSL as a non-differentiable signal in a reinforced manner, promoting better generalization performance.
- We conduct a novel investigation on integration frameworks that combine the power of both RL loss and standard semi-supervised loss, providing a brand new approach that has the potential to lead to more accurate and robust SSL models.
- Extensive experiments demonstrate that our proposed method outperforms state-of-the-art SSL approaches and validate the integration of RL strengths in SSL.

## 2 Related Work

### 2.1 Semi-Supervised Learning

Existing SSL approaches can be broadly classified into three primary categories: regularization-based methods, teacher-student-based methods, and pseudo-labeling techniques.

#### 2.1.1 Regularization-Based Methods

A prevalent research direction in SSL focuses on regularization-based methods, which introduce additional terms to the loss function to promote specific properties of the model. For instance, the Π-model [6] and Temporal-Ensemble [6] incorporate consistency regularization into the loss

function, with the latter employing the exponential moving average of model predictions. Virtual Adversarial Training (VAT) [5] is yet another regularization-based technique that aims to make deep neural networks robust to adversarial perturbations. In a similar vein, Consistency Regularization for Generative Adversarial Networks (CR-GAN) [7] integrates a generative adversarial network (GAN) with a consistency regularization term, facilitating the effective generation of pseudo-labels for unlabeled data.

### 2.1.2 Teacher-Student-Based Methods

Teacher-student-based methods offer an alternative approach in SSL research. These techniques train a student network to align its predictions with those of a teacher network on unlabeled data. Mean Teacher (MT) [8], a prominent example in this category, leverages an exponential moving average (EMA) of successive weights from the student model to obtain the teacher model. To enhance performance, MT + Fast SWA [13] combines Mean Teacher with Fast Stochastic Weight Averaging. Smooth Neighbors on Teacher Graphs (SNTG) [14] takes a different approach, utilizing a graph for the teacher to regulate the distribution of features in unlabeled samples. Meanwhile, Interpolation Consistency Training (ICT) [15] aims to promote consistent predictions across interpolated data points by ensuring that a model's predictions on an interpolated set of unlabeled data points remain consistent with the interpolation of the predictions on those points.

### 2.1.3 Pseudo-Labeling Methods

Pseudo-labeling is an effective way to extend the labeled set when the number of labels is limited. Pseudo-Label [16] produces labels for unlabeled data using model predictions and filters out low-confidence predictions. MixMatch [1] employs data augmentation to create multiple input versions, obtaining predictions for each and averaging them to generate pseudo-labels. In contrast, works such as ReMixMatch [17], UDA [18], and FixMatch [19] apply confidence thresholds to produce pseudo-labels for weakly augmented samples, which subsequently serve as annotations for strongly augmented samples. Label propagation methods, including TSSDL [20] and LPD [21], assign pseudo-labels based on local neighborhood density. DASO [22] combines confidence-based and density-based pseudo-labels in varying ways for each class. Approaches such as Dash [23] and FlexMatch [24] dynamically adjust confidence thresholds in a curriculum learning manner to generate pseudo-labels. Meta Pseudo-Labels [25] uses a bi-level optimization strategy, deriving the teacher update rule from student feedback, to learn from limited labeled data. Co-Training [26] is an early representative of pseudo-lableing which involves training two classifiers on distinct subsets of unlabeled data and using confident predictions to produce pseudo-labels for one another. Similarly, Tri-Training [27] trains three classifiers on separate unlabeled data subsets and generates pseudo-labels based on the disagreements between their predictions.

## 2.2 Reinforcement Learning

Reinforcement Learning (RL) is a field of study that focuses on optimizing an agent's decision-making abilities by maximizing the cumulative reward obtained through interactions with its environment [28]. RL methodology has been widely applied to solve many other learning problems, including searching for optimized network architectures [29], training sequence models for text generation by receiving reward signals [30, 31], and solving online planning problems [32]. Recently, RL has been applied to fine-tune complex models that typically fail to align with users' preferences. Moreover, based on RL from Human Feedback (RLHF; [9–11]), ChatGPT achieves great success in dialogue generation by fine-tuning Large Language Models (LLM) [33]. It frames the training of LLM as a bandit problem, specifically a *one-armed bandit problem* [28], where the objective is to determine the optimal action (dialogue generation) for a given state (user prompt) within a single step, demonstrating the capacity of RL for prediction tasks.

The bandit problem was originally described as a statistical decision model used by agents to optimize their decision-making process [34]. In this problem, an agent receives a reward upon taking an action and learns to make the best decision by maximizing the given reward. The bandit problem found its application in economics and has been widely used in market learning, specifically in finding the optimal market demands or prices to maximize expected profits [35]. Bergemann et al. [36] and Lattimore et al. [37] have extensively discussed the literature and modern applications of the bandit problem. Additionally, Mortazavi et al. [38] introduced a Single-Step Markov Decision Process

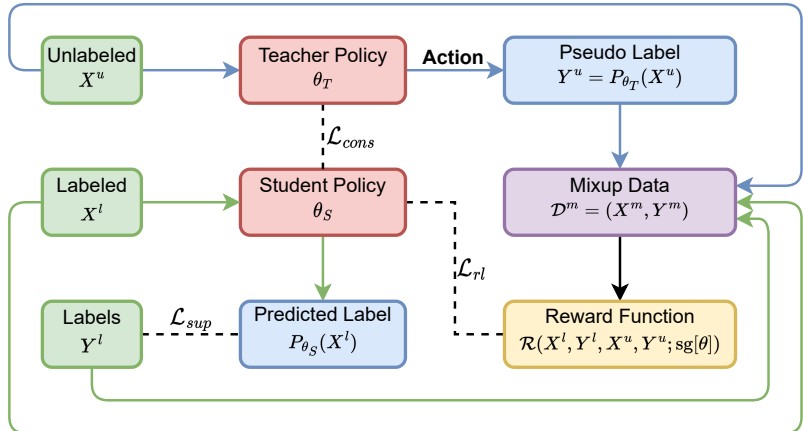

Figure 1: Overview of the RLGSSL Framework. The prediction networks (student $\theta_S$, teacher $\theta_T$) serve as the policy functions, and the soft pseudo-labeling ($P_{\theta_T}(X^u)$) acts as the actions. The model has three loss terms in total: RL loss ($\mathcal{L}_{\mathrm{rl}}$), supervised loss ($\mathcal{L}_{\mathrm{sup}}$), and consistency loss ($\mathcal{L}_{\mathrm{cons}}$). The teacher policy function is used to execute the actions and compute the consistency loss, while the student policy function is used for all other aspects.

(SSMDP) to formulate the bandit problem in a manner that aligns with modern RL techniques. This advancement enables the utilization of standard RL methods on conventional bandit problems.

## 3   The Proposed Method

We consider the following semi-supervised learning setting: the training data consist of a small number of labeled samples, $\mathcal{D}^l = (X^l, Y^l) = \{(x_i^l, \mathbf{y}_i^l)\}_{i=1}^{N^l}$, and a large number of unlabeled samples, $\mathcal{D}^u = X^u = \{x_i^u\}_{i=1}^{N^u}$, with $N^u \gg N^l$, where $x_i^l$ (or $x_i^u$) denotes the input instance and $\mathbf{y}_i^l$ denotes the one-hot label vector with length $C$. The goal is to train a $C$-class classifier $f_\theta : \mathcal{X} \to \mathcal{Y}$ that generalizes well to unseen test data drawn from the same distribution as the training data.

In this section, we present the proposed RLGSSL method, which formulates SSL as a one-armed bandit problem with a continuous action space, and deploys a novel RL loss to guide the SSL process based on a reward function specifically designed for semi-supervised data. Moreover, we further incorporate a semi-supervised teacher-student framework to augment the RL loss with a supervised loss and a prediction consistency regularization loss, aiming to enhance the learning stability and efficacy. Figure 1 illustrates the overall framework of the proposed RLGSSL, and the following subsections will elaborate on the approach.

### 3.1   Reinforcement Learning Formulation for SSL

We treat SSL as a special one-armed bandit problem with a continuous action space. One-armed bandit problem can be considered a single-step Markov Decision Process (MDP) [38]. In this problem, the agent takes a single action and receives a reward based on that action. The state of the environment is not affected by the action. The one-armed bandit problem involves selecting an action to maximize an immediate reward, which can be regarded as learning a policy function under the RL framework. Formulating SSL as a one-armed bandit problem within the RL framework and deploying RL techniques to guide SSL requires defining the following key components: state space $\mathcal{S}$, action space $\mathcal{A}$, a policy function $\pi : \mathcal{S} \to \mathcal{A}$, and a reward function $\mathcal{R} : \mathcal{S} \times \mathcal{A} \to \mathbb{R}$. The objective is to learn an optimal policy $\pi^\star$ that maximizes the expected one-time reward $\mathcal{R}(s, \pi(\cdot|s))$ in the given environment (state $s$): $\pi^\star = \arg\max_\pi \ J_r(\pi) = \sum_a \pi(a|s)\mathcal{R}(s, a)$.

**State**   The state encapsulates the provided knowledge about the environment and is used as input for the policy function. As the action does not affect the state of the environment under one-armed bandit problem, we use the observed data from the SSL problem as the state, i.e., $s = (X^l, Y^l, X^u)$.

**Action and Policy Function** As the goal of SSL is to learn an optimal classifier $f_\theta$ (i.e., prediction network parameterized with $\theta$), we use the classifier $f_\theta$, usually denoted by its parameters $\theta$, as the policy function $\pi_\theta$ to *unify the goals of RL and SSL*. In particular, we consider a probabilistic policy function/prediction network $\pi_\theta(\cdot) = P_\theta(\cdot)$. Since a policy function is used to project a mapping from the state $s$ to the action space, $a = \pi_\theta(\cdot|s)$, by using a probabilistic prediction network as the policy function, it naturally determines a continuous action space $\mathcal{A}$. Specifically, given the fixed state $s$, taking an action is equivalent to making probabilistic predictions on the unlabeled data in $s$: $Y^u = P_\theta(X^u) = \pi_\theta(\cdot|s)$, as the labeled data already has labels. For each unlabeled instance $x_i^u$, the action is a probability vector produced as $\mathbf{y}_i^u = P_\theta(x_i^u)$, which can be regarded as soft pseudo-labels in the SSL setting. This links the action of RL to the pseudo-labeling in SSL.

### 3.1.1 Reward Function

The reward function serves as feedback to evaluate the performance of the action (prediction) provided by the policy. It needs to be thoughtfully crafted to maximize the model's ability to extract useful information from both labeled and unlabeled data, which is central to the SSL paradigm. The underlying motivation is to guide the learning process to induce a more generalizable and robust prediction model. To this end, we adopt a data mixup [39] strategy to produce new data points from the given labeled data $(X^l, Y^l)$ and pseudo-labeled data $(X^u, Y^u)$, which together form the state-action pair $(s, a)$, through linear data interpolation, and assess the prediction model's generalization ability on such data points as the reward signal. This decision is inspired by the proven effectiveness of mixup in enhancing model performance in various tasks. The idea of data mixup is to generate virtual training examples by creating convex combinations of pairs of input data and their corresponding labels. This technique encourages the model to learn more fluid decision boundaries, leading to improved generalization capabilities.

Specifically, we propose to generate new data points by performing *inter-mixup* between labeled and unlabeled data points, aiming to maintain a balanced utilization of both labeled and unlabeled data. In order to address the size discrepancy between the labeled dataset $\mathcal{D}^l$ and the unlabeled dataset $\mathcal{D}^u$ with $N^u \gg N^l$, we replicate the labeled dataset $\mathcal{D}^l$ by a factor of $r = \lceil \frac{N^u}{N^l} \rceil$ times, resulting in an extended labeled dataset $\widetilde{\mathcal{D}}^l$. After shuffling the data points in each set, we generate a mixup data point by mixing an unlabeled point $x_i^u \in \mathcal{D}^u$ with a labeled point $x_i^l \in \widehat{\mathcal{D}}^l$ along with their corresponding pseudo-label $\mathbf{y}_i^u \in \mathcal{D}^u$ and label $\mathbf{y}_i^l \in \widetilde{\mathcal{D}}^l$:

$$x_i^{\mathrm{m}} = \mu\, x_i^u + (1-\mu)\, x_i^l, \qquad \mathbf{y}_i^{\mathrm{m}} = \mu\, \mathbf{y}_i^u + (1-\mu)\, \mathbf{y}_i^l \tag{1}$$

where the mixing parameter $\mu$ is sampled from a Beta distribution, Beta(1,1). With this procedure, we can generate $N^{\mathrm{m}} = N^u$ mixup samples by mixing all the unlabeled samples with the samples in the extended labeled set.

We then define the reward function to measure the negative mean squared error (MSE) between the model's prediction $P_\theta(x_i^{\mathrm{m}})$ and the mixup label $\mathbf{y}_i^{\mathrm{m}}$ for each instance in the mixup set. This results in a single, comprehensive metric that quantifies the overall negative disagreement between the model's predictions and the mixup labels over a large set of interpolated data points:

$$\mathcal{R}(s, a; \mathrm{sg}[\theta]) = \mathcal{R}(X^l, Y^l, X^u, Y^u; \mathrm{sg}[\theta]) = -\frac{1}{C \cdot N^{\mathrm{m}}} \sum_{i=1}^{N^{\mathrm{m}}} ||P_\theta(x_i^{\mathrm{m}}) - \mathbf{y}_i^{\mathrm{m}}||_2^2 \tag{2}$$

where $C$ denotes the number of classes and $\mathrm{sg}[\cdot]$ is the stop gradient operator which stops the flow of gradients during the backpropagation process. This ensures that the reward function is solely employed for model assessment, rather than being directly utilized for model updating, enforcing the working mechanisms of RL. Mixup labels capture both the supervision information in the labeled data and the uncertainty in the pseudo-labels of unlabeled data. With the designed reward function, a good reward value can only be returned when the prediction model not only exhibits strong alignment with the labeled data but also delivers accurate predictions on the unlabeled data. Consequently, through RL, this reward function will not only promote accurate predictions but also enhance the model's robustness and generalizability.

### 3.1.2 Reinforcement Learning Loss

By deploying the probabilistic predictions on the unlabeled data, $Y^u = \pi_\theta(X^u) = P_\theta(X^u)$, as the action, we adopt a deterministic policy. Following the principle of one-armed bandit problem on

maximizing the expected one-time reward w.r.t. the policy output, we introduce a weighted negative reward based on the deterministic policy's output as the RL loss for the proposed RLGSSL, thereby exploiting non-differentiable reward signals while enabling policy gradient with a deterministic policy. Specifically, we treat the output of the policy network, $Y^u$, as a uniform distribution over the set of $N^u$ probability vectors, $\{\mathbf{y}_1^u, \cdots, \mathbf{y}_{N^u}^u\}$, predicted for the unlabeled instances. Let $\mathbf{e} = \mathbf{1}/C$ denote a discrete uniform distribution vector with length $C$—the number of classes. We design the following KL-divergence weighted negative reward as the RL loss:

$$\mathcal{L}_{\text{rl}}(\theta) = -\mathbb{E}_{\mathbf{y}_i^u \sim \pi_\theta} \text{KL}(\mathbf{e}, \mathbf{y}_i^u) \mathcal{R}(s, a; \text{sg}[\theta]) = -\mathbb{E}_{x_i^u \in \mathcal{D}_u} \text{KL}(\mathbf{e}, P_\theta(x_i^u)) \mathcal{R}(s, a; \text{sg}[\theta]) \quad (3)$$

where the KL-divergence term measures the distance of each label prediction probability vector $\mathbf{y}_i^u$ from a uniform distribution vector; $\mathcal{R}(s, a; \text{sg}[\theta_S])$ is treated as a non-differentiable reward function. Given that a uniform probability distribution signifies the least informative prediction outcome, the expected KL-divergence captures the level of informativeness in the policy output and hence serves as a meaningful weight for the reward, which inherently encourages the predictions to exhibit greater discrimination.

The minimization of this loss function over the prediction network parameterized by $\theta$ is equivalent to learning an optimal policy function $\pi_\theta$ by maximizing the KL-divergence weighted reward, which aims at an optimal policy function (also the probabilistic classifier $P_\theta$) that not only maximizes the reward signal but is also discriminative. From the perspective of SSL, the utilization of this novel RL loss introduces a fresh approach to designing prediction loss functions. Instead of directly optimizing the alignment between predictions and targets, it offers a gradual learning process guided by reward signals. This innovative approach presents a more adaptive and flexible solution for complex data scenarios, where traditional optimization-based methods may fall short.

## 3.2  Teacher-Student Framework

Teacher-student models [8] have been popularly deployed to exploit unlabeled data for SSL, improving the learning stability. We extend this mechanism to provide a teacher-student framework for RL-guided SSL by maintaining a dual set of model parameters: the student policy/model parameters $\theta_S$, and the teacher policy/model parameters $\theta_T$. The student model is directly updated through training, whereas the teacher model is updated via an exponential moving average (EMA) of the student model. The update is conducted as follows:

$$\theta_T = \beta \theta_T + (1 - \beta) \theta_S \quad (4)$$

where $\beta$ denotes a hyperparameter that modulates the EMA's decay rate. The utilization of the EMA update method ensures a stable and smooth transfer of knowledge from the student model to the teacher model. This leads to a teacher model with consistent and reliable parameter values that are not susceptible to random or erratic fluctuations during the training process. Leveraging this desirable characteristic, we propose to employ the *teacher* model for *executing actions* within the RL framework described in the section 3.1 above; that is, $Y^u = P_{\theta_T}(X^u)$, while retaining the *student* model for *other aspects*. By doing so, we ensure that stable actions are taken, reducing the impact of random noise in the pseudo-labels and enhancing the accuracy of reward evaluation and reinforcement strength.

Within the teacher-student framework, we further propose to augment the RL loss with a supervised loss $\mathcal{L}_{\text{sup}}$ on the labeled data and a consistency regularization loss $\mathcal{L}_{\text{cons}}$ on the unlabeled data. We adopt a standard cross-entropy loss function $\ell_{CE}$ to compute the supervised loss, promoting accurate predictions on $\mathcal{D}^l$ where the ground-truth labels are available:

$$\mathcal{L}_{\text{sup}}(\theta_S) = \mathbb{E}_{(x^l, \mathbf{y}^l) \in \mathcal{D}^l} \left[ \ell_{CE} \left( P_{\theta_S}(x^l), \mathbf{y}^l \right) \right] \quad (5)$$

This loss can enhance effective exploitation of the ground-truth label information, providing a solid basis for exploring the parameter space via RL. The consistency loss $\mathcal{L}_{\text{cons}}$ is deployed to encourage prediction consistency between the student and teacher models on the unlabeled data $\mathcal{D}^u$:

$$\mathcal{L}_{\text{cons}}(\theta_S) = \mathbb{E}_{x^u \in \mathcal{D}^u} \left[ \text{KL} \left( P_{\theta_S}(x^u), P_{\theta_T}(x^u) \right) \right] \quad (6)$$

where $\text{KL}(\cdot, \cdot)$ denotes the Kullback-Leibler divergence between two probability distributions. By enforcing consistency, this loss encourages the student model to make more confident and reliable predictions, reducing the impact of random or misleading information in the training set. It also acts as a form of regularization, discouraging the student model from overfitting the labeled data.

**Algorithm 1** Pseudo-Label Based Policy Gradient Descent

---

**Input:** $\mathcal{D}^l, \mathcal{D}^u$, and extended $\widetilde{\mathcal{D}}^l$;    initialized $\theta_S, \theta_T$;    hyperparameters

**for** $iteration = 1$ to maxiters **do**
    **for** $x_i^u \in \mathcal{D}^u$ **do**
        Compute soft pseudo-label vector $\mathbf{y}_i^u = P_{\theta_T}(x_i^u)$ to form $(x_i^u, \mathbf{y}_i^u)$
    **end for**
    Generate mixup data $\mathcal{D}^m = (X^m, Y^m)$ on $\mathcal{D}^u$ and $\widetilde{\mathcal{D}}^l$ using Eq.(1) with shuffling
    **for** $step = 1$ to maxsteps **do**
        Draw a batch of data $B = \{(x_i^m, \mathbf{y}_i^m)\}$ from $\mathcal{D}^m$
        Calculate the reward function $\mathcal{R}(\cdot; sg[\theta_S])$ using the batch $B$
        Compute the objective in Eq.(7)
        Update the policy parameters $\theta_S$ via gradient descent
        Update teacher model $\theta_T$ via EMA in Eq.(4)
    **end for**
**end for**

---

Table 1: Performance of RLGSSL and state-of-the-art SSL algorithms with the CNN-13 network. We report the average test errors and the standard deviations of 5 trials.

| Dataset | CIFAR-10 | | | CIFAR-100 | |
|---|---|---|---|---|---|
| Number of Labeled Samples | 1000 | 2000 | 4000 | 4000 | 10000 |
| Supervised | $39.95_{(0.75)}$ | $27.67_{(0.12)}$ | $20.42_{(0.21)}$ | $58.31_{(0.89)}$ | $44.56_{(0.30)}$ |
| Supervised + MixUp [39] | $31.83_{(0.65)}$ | $24.22_{(0.15)}$ | $17.37_{(0.35)}$ | $54.87_{(0.07)}$ | $40.97_{(0.47)}$ |
| Π-model [6] | $28.74_{(0.48)}$ | $17.57_{(0.44)}$ | $12.36_{(0.17)}$ | $55.39_{(0.55)}$ | $38.06_{(0.37)}$ |
| Temp-ensemble [6] | $25.15_{(1.46)}$ | $15.78_{(0.44)}$ | $11.90_{(0.25)}$ | - | $38.65_{(0.51)}$ |
| Mean Teacher[8] | $21.55_{(0.53)}$ | $15.73_{(0.31)}$ | $12.31_{(0.28)}$ | $45.36_{(0.49)}$ | $35.96_{(0.77)}$ |
| VAT [5] | $18.12_{(0.82)}$ | $13.93_{(0.33)}$ | $11.10_{(0.24)}$ | - | - |
| SNTG [14] | $18.41_{(0.52)}$ | $13.64_{(0.32)}$ | $10.93_{(0.14)}$ | - | $37.97_{(0.29)}$ |
| Learning to Reweight [40] | $11.74_{(0.12)}$ | - | $9.44_{(0.17)}$ | $46.62_{(0.29)}$ | $37.31_{(0.47)}$ |
| MT + Fast SWA [13] | $15.58_{(0.12)}$ | $11.02_{(0.23)}$ | $9.05_{(0.21)}$ | - | $33.62_{(0.54)}$ |
| ICT [15] | $12.44_{(0.57)}$ | $8.69_{(0.15)}$ | $7.18_{(0.24)}$ | $40.07_{(0.38)}$ | $32.24_{(0.16)}$ |
| RLGSSL (Ours) | $\mathbf{9.15}_{(0.57)}$ | $\mathbf{6.90}_{(0.11)}$ | $\mathbf{6.11}_{(0.10)}$ | $\mathbf{36.92}_{(0.45)}$ | $\mathbf{29.12}_{(0.20)}$ |

### 3.3 Training Algorithm for RL-Guided SSL

The learning objective for the RLGSSL approach is formed by combining the reinforcement learning loss $\mathcal{L}_{\mathrm{rl}}$ with the two augmenting loss terms, the supervised loss $\mathcal{L}_{\mathrm{sup}}$ and the consistency loss $\mathcal{L}_{\mathrm{cons}}$, using hyperparameters $\lambda_1$ and $\lambda_2$:

$$\mathcal{L}(\theta_S) = \mathcal{L}_{\mathrm{rl}} + \lambda_1 \mathcal{L}_{\mathrm{sup}} + \lambda_2 \mathcal{L}_{\mathrm{cons}} \tag{7}$$

By deploying such a joint loss, the RLGSSL framework can benefit from the strengths of both reinforcement exploration and semi-supervised learning. The RL component, in particular, introduces a dynamic aspect to the learning process, enabling the model to improve iteratively based on its own experiences. This innovative combination of losses allows the model to effectively learn from limited labeled data while still exploiting the abundance of unlabeled data.

We develop a stochastic batch-wise gradient descent algorithm to minimize the joint objective in Eq. (7) for RL-guided semi-supervised training. The procedure of this algorithm is summarized in Algorithm 1.

## 4 Experiments

### 4.1 Experimental Setup

**Datasets** We conducted comprehensive experiments on four image classification benchmarks: CIFAR-10, CIFAR-100 [43], SVHN [44], and STL-10 [45]. We adhere to the conventional dataset splits used in the literature. Consistent with previous works, on each dataset we preserved the labels of a randomly selected subset of training samples with an equal number of samples for each class, and left the remaining samples unlabeled. In order to compare with previous works in the same settings,

Table 2: Performance of RLGSSL and state-of-the-art SSL algorithms with the CNN-13 network. We report the average test errors and the standard deviations of 5 trials.

| | VAT [5] | Π-model [6] | Temp-ensemble [6] | MT [8] | ICT [15] | SNTG [14] | RLGSSL (Ours) |
|---|---|---|---|---|---|---|---|
| SVHN/500 | - | $6.65_{(0.53)}$ | $5.12_{(0.13)}$ | $4.18_{(0.27)}$ | $4.23_{(0.15)}$ | $3.99_{(0.24)}$ | $\mathbf{3.12}_{(0.07)}$ |
| SVHN/1000 | $5.42_{(0.00)}$ | $4.82_{(0.17)}$ | $4.42_{(0.16)}$ | $3.95_{(0.19)}$ | $3.89_{(0.04)}$ | $3.86_{(0.27)}$ | $\mathbf{3.05}_{(0.04)}$ |

Table 3: Comparison results in terms of mean test error and standard deviation using WRN-28-2 as the backbone on CIFAR-10 and SVHN and using WRN-28-8 as the backbone on CIFAR-100.

| Dataset | CIFAR-10 | | | CIFAR-100 | | SVHN |
|---|---|---|---|---|---|---|
| Number of Labeled Samples | 250 | 1000 | 4000 | 2500 | 10000 | 1000 |
| Mean Teacher [8] | $32.32_{(2.30)}$ | $17.32_{(4.00)}$ | $10.36_{(0.25)}$ | $53.91_{(0.57)}$ | $35.83_{(0.24)}$ | $5.65_{(0.45)}$ |
| ICT [15] | - | - | $7.66_{(0.17)}$ | - | - | $3.53_{(0.07)}$ |
| MixMatch [1] | $11.05_{(0.15)}$ | $7.75_{(0.32)}$ | $6.24_{(0.06)}$ | $39.94_{(0.37)}$ | $28.31_{(0.33)}$ | $3.27_{(0.31)}$ |
| ReMixMatch [17] | $5.44_{(0.05)}$ | $6.27_{(0.34)}$ | $6.24_{(0.06)}$ | $27.14_{(0.23)}$ | $23.78_{(0.12)}$ | $3.27_{(0.31)}$ |
| FixMatch [19] | $5.07_{(0.35)}$ | - | $4.26_{(0.05)}$ | $28.29_{(0.11)}$ | $22.60_{(0.12)}$ | $2.28_{(0.11)}$ |
| Meta-Semi [41] | - | $7.34_{(0.22)}$ | $6.10_{(0.10)}$ | - | - | - |
| Meta Pseudo-Labels [25] | - | - | $3.89_{(0.07)}$ | - | - | $1.99_{(0.07)}$ |
| MarginMatch [42] | $\mathbf{4.73}_{(0.12)}$ | - | $3.98_{(0.02)}$ | $23.71_{(0.13)}$ | $21.39_{(0.12)}$ | $1.93_{(0.01)}$ |
| RLGSSL (Ours) | $5.01_{(0.27)}$ | $\mathbf{4.92}_{(0.25)}$ | $\mathbf{3.52}_{(0.06)}$ | $\mathbf{23.18}_{(0.43)}$ | $\mathbf{20.15}_{(0.34)}$ | $\mathbf{1.92}_{(0.05)}$ |

we performed experiments on CIFAR-10 with various numbers ($N^l \in \{250, 1,000, 2,000, 4,000\}$) of labeled samples, on CIFAR-100 with 2,500, 10,000 and 4,000 labeled samples, on SVHN with 1,000 and 500 labeled samples, and on STL-10 with 1,000 labeled images.

**Implementation Details**   We conducted experiments using four different network architectures used in the literature: a 13-layer Convolutional Neural Network (CNN-13), a Wide-Residual Network with 28 layers and a widening factor of 2 (WRN-28-2), a Wide-RestNet-28-8 (WRN-28-8) and a Wide-RestNet-37-2 (WRN-37-2). For training the CNN-13 architecture, we employed the SGD optimizer with a Nesterov momentum of 0.9. We used an L2 regularization coefficient of 1e-4 for CIFAR-10 and CIFAR-100, and 5e-5 for SVHN. The initial learning rate was set to 0.1, and the cosine learning rate annealing technique proposed in previous studies [46, 15] was utilized. For the WRN-28-2 architecture, we followed the suggestion from MixMatch [1] and used an L2 regularization coefficient of 4e-4. For WRN-37-2, the training configuration includes the SGD optimizer, an L2 regularization coefficient of 5e-4, and an initial learning rate of 0.01. Finally, the training configuration for the WRN-28-8 model includes using the SGD optimizer, an L2 regularization coefficient of 0.001, and starting with a learning rate of 0.01. To compute the parameters of the teacher model, we employed the EMA method with a decay rate $\beta = 0.999$. We selected all hyperparameters and training techniques based on relevant studies to ensure a fair comparison between our approach and the existing methods. Specifically for RLGSSL, we set the batch size to 128, and set $\lambda_1 = \lambda_2 = 0.1$. We first pre-train the model for 50 epochs using the Mean-Teacher algorithm and then proceed to the training procedure of RLGSSL for 400 epochs. We ran each experiment five times and reported the mean test errors with their standard deviations.

## 4.2   Comparison Results

We compare RLGSSL with a great number of SSL algorithms, including Supervised + MixUp [39], Π-model [6], Temp-ensemble [6], Mean Teacher [8], VAT [5], SNTG [14], Learning to Reweight [40], MT + Fast SWA [13], MixMatch [1], ReMixMatch [17], FixMatch [19], MarginMatch [42], Meta-Semi [41], Meta Pseudo-Labels [25], and ICT [15], using CNN-13, WRN-28-2, WRN-28-8 or WRN-37-2 as the backbone network.

Table 1 reports the comparison results on CIFAR-10 with 4,000, 2,000, and 1,000 labeled samples and on CIFAR-100 with 10,000 and 4,000 labeled samples when CNN-13 is used as the backbone network. On CIFAR-10, RLGSSL outperforms all the other compared methods across all settings with different numbers of labeled samples. With 1,000 labeled samples, RLGSSL surpasses the second best method, *Learning to Reweight*, by a significant margin of $2.59\%$ on CIFAR-10, achieving an average test error of $9.15\%$. This pattern of outperformance continues with different numbers (2,000 and 4,000) of labeled samples, where RLGSSL yields lowest test error rates and outperforms ICT—the second best method—by a margin of $1.79\%$ and $1.07\%$ respectively. The results on the

Table 4: Comparison results in terms of mean test error and standard deviation using WRN-37-2 as the backbone on STL-10.

|  | Π Model [6] | MeanTeacher [8] | MixMatch [1] | UDA [2] | RLGSSL (Ours) |
|---|---|---|---|---|---|
| STL-10 / 1000 | $26.23_{(0.82)}$ | $21.43_{(2.39)}$ | $10.41_{(0.61)}$ | $7.66_{(0.56)}$ | $\mathbf{6.12}_{(0.52)}$ |

Table 5: Ablation study results on CIFAR-100 using 10000 and 4000 labels with CNN-13 as the backbone. The average test errors and standard deviations over 5 trials are reported.

|  | RLGSSL | $-$w/o $\mathcal{L}_{\mathrm{rl}}$ | $-$w/o $\mathcal{L}_{\mathrm{sup}}$ | $-$w/o $\mathcal{L}_{\mathrm{cons}}$ | $-$w/o EMA | $-$w/o mixup |
|---|---|---|---|---|---|---|
| CIFAR-100/4000 | $\mathbf{36.92}_{(0.45)}$ | $44.92_{(0.55)}$ | $39.52_{(0.58)}$ | $38.78_{(0.48)}$ | $43.12_{(0.52)}$ | $40.12_{(0.51)}$ |
| CIFAR-100/10000 | $\mathbf{29.12}_{(0.20)}$ | $33.12_{(0.52)}$ | $32.67_{(0.45)}$ | $31.48_{(0.32)}$ | $32.84_{(0.45)}$ | $31.48_{(0.32)}$ |
|  | RLGSSL | $\mathcal{R}=1$ | $\mathcal{R}:\mu=0$ | $\mathcal{R}(\mathrm{MSE}\rightarrow\mathrm{KL})$ | $\mathcal{R}(\mathrm{MSE}\rightarrow\mathrm{JS})$ | $\mathcal{R}:$ w/o sg$[\theta]$ |
| CIFAR-100/4000 | $\mathbf{36.92}_{(0.45)}$ | $39.52_{(0.63)}$ | $39.54_{(0.33)}$ | $38.02_{(0.42)}$ | $39.52_{(0.45)}$ | $40.62_{(0.55)}$ |
| CIFAR-100/10000 | $\mathbf{29.12}_{(0.20)}$ | $31.25_{(0.62)}$ | $32.37_{(0.57)}$ | $31.12_{(0.52)}$ | $31.39_{(0.68)}$ | $32.12_{(0.62)}$ |

CIFAR-100 dataset are similarly impressive. For 4,000 labeled samples, RLGSSL again outperforms ICT, the second best method, with a margin of 3.15%. As the number of labeled samples escalates to 10,000, RLGSSL maintains its performance edge, outperforming the second best method ICT by a margin of 3.12%.

Table 2 reports the comparison results on the SVHN dataset with CNN-13 as the backbone network. Our method, RLGSSL, surpasses all the other compared SSL techniques for both settings. Specifically, for 500 labeled samples, RLGSSL achieves the lowest test error of 3.12%, which is 0.87% lower than the second-best method, SNTG. For 1,000 labeled samples, RLGSSL also shows superior performance with a test error of 3.05%, outperforming the second-best method, SNTG, by 0.81%.

Table 3 presents the comparative outcomes across three datasets, utilizing WRN-28-2 as the backbone network for CIFAR-10 and SVHN, and WRN-28-8 for CIFAR-100. On CIFAR-10, with the number of labeled samples increasing from 250 to 4,000, RLGSSL showed better performance compared to the other methods in most cases. For 1,000 labeled samples, our method improved over the best competing method, ReMixMatch, by a margin of 1.35%. Additionally, for 4,000 labeled samples RLGSSL outperforms *Meta Pseudo-labels* which is the second best method. On CIFAR-100, with 2,500 and 10,000 labeled samples, RLGSSL surpassed the state-of-the-art MarginMatch. Moreover, on SVHN with 1,000 labeled samples, RLGSSL also outperformed the MarginMatch. This confirms the robustness of RLGSSL across various settings, even when the labeled data is limited in quantity.

Table 4 presents the comparison results of various SSL methods on the STL-10 dataset, utilizing WRN-37-2 as the backbone network. With a fixed number of 1,000 labeled samples, RLGSSL achieves remarkable performance with a mean test error of 6.12%, which outperforms previous state-of-the-art methods, including MixMatch and UDA, showcasing the effectiveness of our RLGSSL.

### 4.3  Ablation Study

In order to evaluate the significance of various components of our RLGSSL approach, we conducted an ablation study on the CIFAR-100 dataset using the CNN-13 network. In particular, we compared the full model RLGSSL with the following variants: (1) "$-$w/o $\mathcal{L}_{\mathrm{rl}}$", which drops the RL loss $\mathcal{L}_{\mathrm{rl}}$; (2) "$-$w/o $\mathcal{L}_{\mathrm{sup}}$", which excludes the supervised loss; (3) "$-$w/o $\mathcal{L}_{\mathrm{cons}}$", which does not include the consistency loss; (4) "$-$w/o EMA", which drops the teacher model by disabling the EMA update; and (5) "$-$w/o mixup", which only uses unlabelled data in the reward function ($\mu = 1$), and the mixup operation is excluded. The ablation results are reported in the top section of Table 5. The full model, RLGSSL, achieved the lowest test errors, confirming the overall effectiveness of our method. The most significant observation from our study lies in the removal of the RL loss $\mathcal{L}_{\mathrm{rl}}$. Upon removal of this component, we witness a substantial increase in test errors, which highlights the indispensable role played by the RL component in our model. The ablation study further illustrates the importance of each component by analyzing the performance of the model when the component is removed. In each of these cases, we observe that the removal of any individual component consistently leads to an increase in test errors. This finding underpins the notion that each component of the RLGSSL model plays a significant role in the overall performance of the model.

In addition, we also conducted another set of ablation study centered on the proposed RL loss and the reward function. We compared the full model RLGSSL with the following variants: (1) "$\mathcal{R} = 1$",

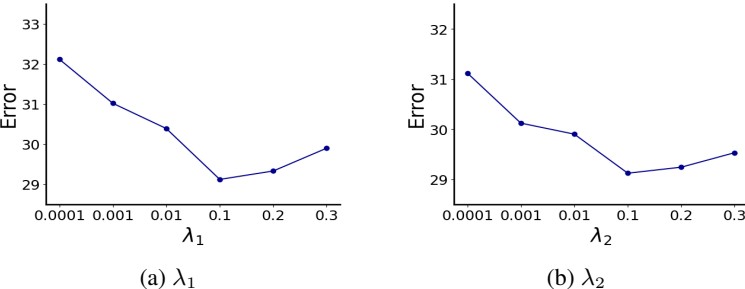

Figure 2: Sensitivity analysis for hyper-parameters $\lambda_1$ and $\lambda_2$ on CIFAR-100 using 10000 labeled samples.

which drops RL by setting the reward as a constant 1; (2) "$\mathcal{R} : \mu = 0$", which only uses labeled data to compute the reward by setting $\mu = 0$; (3) "$\mathcal{R}(\text{MSE} \to \text{KL})$", which replaces the mean squared error of the reward function in Eq. (2) with the KL-divergence loss; (4) "$\mathcal{R}(\text{MSE} \to \text{JS})$", which replaces the mean squared error of the reward function in Eq. (2) with the JS-divergence loss; and (5) "$\mathcal{R} : \text{w/o sg}[\theta]$", which removes the stop-gradient operator from the reward function and makes the reward function differentiable w.r.t $\theta$. The ablation results are reported in the bottom section of Table 5. We can see that all these variants with altered reward functions produced degraded performance comparing to the full model with the proposed reward function. In particular, the performance degradation of "$\mathcal{R} = 1$" and "$\mathcal{R} : \text{w/o sg}[\theta]$" that drop RL in different manners further validates the contribution of the proposed framework of guiding SSL with RL.

## 4.4 Hyper-parameter Analysis

We conduct sensitivity analysis over the two hyper-parameters of the proposed RLGSSL: $\lambda_1$—the trade-off parameter for the supervised loss, and $\lambda_2$—the trade-off parameter for the consistency loss. The results are reported in Figure 2. In the case of $\lambda_1$, lower values (e.g., $1e$-4 and $1e$-3) result in less emphasis on the supervised loss term in the objective function. As a result, the model might not learn effectively from the limited available labeled data, leading to increased test error rates. Conversely, higher values of $\lambda_1$ (e.g., $0.2$ and $0.3$) may overemphasize the supervised loss term, potentially causing the model to overfit the labeled data and ignore useful information from the unlabeled data. The sweet spot lies in the middle (around $0.1$), attaining a balance between learning from labeled data and leveraging information from unlabeled data. Regarding $\lambda_2$, very low values (e.g., $1e$-4 and $1e$-3) may not enforce sufficient consistency in the model predictions on unlabeled data, resulting in a model that fails to generalize well. However, if $\lambda_2$ is too high (e.g., $0.2$ and $0.3$), the model may overemphasize the consistency constraint, possibly leading to a model that is too rigid to capture the diverse patterns in the data. An optimal value of $\lambda_2$ (around $0.1$ in our experiments) ensures a good balance between encouraging prediction consistency and allowing the model to adapt to the diverse patterns in the data. The optimal value choice for hyperparameters $\lambda_1$ and $\lambda_2$ (around $0.1$) also validates that the RL loss is the main leading term, while the supervised loss and consistency loss are augmenting terms.

## 5 Conclusion

In this paper, we presented Reinforcement Learning-Guided Semi-Supervised Learning (RLGSSL), a unique approach that integrates the principles of RL to tackle the challenges inherent in SSL. This initiative was largely driven by the limitations of conventional SSL techniques. RLGSSL employs a distinctive strategy where an RL-optimized reward function is utilized. This function adaptively promotes better generalization performance through more effectively leveraging both labeled and unlabeled data. We also further incorporated a student-teacher framework to integrate the strengths of RL and SSL. Extensive evaluations were conducted on multiple benchmark datasets, comparing RLGSSL to existing state-of-the-art SSL techniques, and various ablation variants. RLGSSL consistently outperformed these other techniques across all the datasets, which attests to the effectiveness and generalizability of our approach. The results underline the potential of integrating RL principles into SSL, and the RLGSSL method introduced in this paper is a significant stride in this direction.

## Acknowledgement and Disclosure of Funding

This research was supported in part by an NSERC Discovery Grant, the Canada Research Chairs Program, and the Canada CIFAR AI Chairs Program.

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

## A  Computer Resources

Our experiments were performed on setups featuring CPUs with 8 Intel Core processors and 64 GB of RAM. For graphics processing units, we utilized NVIDIA GeForce RTX 3060 cards, each offering 12 GB of VRAM.

## B  Limitation

While our RLGSSL method demonstrates significant improvements in semi-supervised learning tasks, we acknowledge certain limitations that can be addressed in future work. Our current formulation assumes that the unlabeled data is drawn from the same distribution as the labeled data. This assumption might not hold in certain real-world scenarios where labeled and unlabeled data come from different, albeit related, distributions. Future extensions could consider a domain adaptation strategy to handle such scenarios.

Despite these limitations, the proposed work provides a promising direction for integrating reinforcement learning with semi-supervised learning, paving the way for more adaptive and versatile machine learning algorithms.

## C  Broader Impacts

The proposed Reinforcement Learning Guided Semi-Supervised Learning (RLGSSL) method has significant positive social impacts, particularly in access to advanced machine learning techniques and enabling more efficient use of data resources. By effectively leveraging both labeled and unlabeled data, RLGSSL can reduce the dependency on extensive and expensive labeled datasets, making high-performing machine learning models more accessible to organizations with limited labeling resources. This can particularly benefit fields such as healthcare, education, and environmental monitoring, where acquiring labeled data can be challenging and costly.

