# OpenReview forum: "Reinforcement Learning Guided Semi-Supervised Learning"
_NeurIPS.cc/2024/Conference — NeurIPS 2024 poster_

### Official Review · Reviewer_mW6a · 2024-06-17

**Soundness:** 2
**Presentation:** 2
**Contribution:** 2
**Rating:** 3
**Confidence:** 4

**Summary:**

This paper introduces Reinforcement Learning-Guided Semi-Supervised Learning (RLGSSL), a novel method that combines reinforcement learning (RL) with semi-supervised learning (SSL). By formulating SSL as a one-armed bandit problem, the authors employ a RL-based loss function to guide the learning process. Additionally, the method incorporates a semi-supervised teacher-student framework to improve learning stability.

**Strengths:**

The paper introduces a unique approach by framing semi-supervised learning as a one-armed bandit problem and integrating reinforcement learning to optimize the generation of pseudo-labels.

**Weaknesses:**

The rationality of the method design is not entirely convincing. Firstly, in Equation (2), why is the non-differentiable MSE used instead of the differentiable cross-entropy loss? Why use RL when supervised learning could be applied? Additionally, in the explanation of Equation (3), how is the "meaningful weight" defined? Would this design still be applicable when there is a significant imbalance in the number of samples for each class in the dataset?

Training with RL methods is challenging and often unstable, though RL has its own advantages. The authors are advised to add experiments demonstrating the benefits of the RL-based design compared to a supervised design.

Some descriptions in the paper might be confusing for readers without background knowledge. For example, the terms "convex combination" in line 190 and "fluid decision boundaries" in line 191 might be unclear.

According to the results in Table 3, the improvements brought by the proposed method are not significant.

**Questions:**

1. In line 215, the authors mention the reward function that can enhance the model's robustness and generalization. How is this proven in the experiments?

2. Could the authors further explain the description from lines 228 to 231?

**Limitations:**

In addition to the limitations discussed by the authors in the appendix, please refer to my comments in the Weaknesses section for further details.

---

> ### Author Rebuttal · Authors · 2024-08-06
>
> We sincerely appreciate the time and effort the reviewer has dedicated to reviewing our work.
>
>
> **About the rationality and advantage of using RL**: We use Reinforcement Learning (RL) to enhance the exploration of pseudo-labels. Traditional approaches in Semi-Supervised Learning (SSL) can encounter issues like overfitting and challenges in generating pseudo-labels. RL brings benefits of exploration and the ability to handle non-differentiable operations by treating the predictor as a policy function. Our proposed method establishes a pioneer and novel investigation framework for integrating the power of RL through non-differentiable losses with standard SSL loss. We conducted an ablation study in Table 5 of our manuscript by dropping the RL which is the variant “$\mathcal{R}:$ w/o sg[$\theta$]". This variant demonstrates poorer results compared to our RLGSSL method showcasing the necessity of the bandit framework in our work.
>
>
> **About MSE loss**: We opted for the Mean Squared Error (MSE) when dealing with mixup data in the reward function, aligning with the established practices found in seminal works such as MixMatch [1], which utilizes L2 loss for this purpose.MSE loss is less sensitive to label noise compared to CE loss. Since pseudo-labels for the unlabeled data are not ground truth labels, they may contain some degree of uncertainty or noise, and hence using MSE loss can be a good option.  In addition, a non-differentiable reward function ensures that the reward function is solely employed for model assessment, rather than being directly utilized for model updating, enforcing the working mechanisms of RL.
>
>
> **About imbalance scenarios**:  In this work, we followed the established convention for research in classic semi-supervised learning (SSL). While our method has demonstrated promising results in this context, the specific scenario of class imbalance in the dataset was not directly investigated. The RL component's focus on decision-making and optimization is broadly applicable across different types of learning tasks. Therefore, while specific adjustments might be necessary, the foundational idea of RLGSSL could be extended and applied to different domains and tasks including but not limited to imbalance scenarios. Exploring the applicability and potential adaptations of our approach for imbalanced datasets presents a valuable direction for future work.
>
> **About experiments showing the advantage of RL**:  We conducted an ablation study in Table 6 of our manuscript with a variant labeled "$\mathcal{R}:$ w/o sg[$\theta$]", where we removed the stop-gradient operator from the reward function. This modification allowed the reward function to become differentiable with respect to $\theta$, effectively transforming the setup into a supervised learning framework. This variant showed inferior performance compared to our original RLGSSL method which demonstrates the advantage of using RL.
>
> **About background knowledge**:  Given the constraints of space and the expected familiarity of NeurIPS audiences with standard machine learning terminology, we focused on detailing our novel method rather than explaining well-established concepts such as "convex combination" and "fluid decision boundaries." We appreciate your understanding and encourage readers seeking definitions of these common terms to consult foundational ML literature or resources.
>
> **About improvement**: The improvement offered by our RLGSSL method is substantial as it significantly outperforms state-of-the-art works in semi-supervised learning (SSL). Specifically, RLGSSL surpasses the Interpolation Consistency Training (ICT) by a notable margin of 3.29% on CIFAR10 using just 1,000 labels, as detailed in Table 1 of our manuscript. Similarly, for CIFAR100 with 4,000 labeled instances, the improvement margin is 3.15%. it's noteworthy that our method in Table 3 has already achieved a very low error rate of 3.52%  for CIFAR10 using 4000 labels, where any additional improvement is inherently limited. Note that MarginMatch only achieves a 0.09% improvement over the previous Meta Pseudo-Labels method on the same dataset setting, which is much smaller than the performance gain our method achieved with an even lower error rate.
>
>
> **About effectiveness of the reward function** In the ablation studies presented in Table 5 of our manuscript, we examined the variation where "$\mathcal{R}=1$", effectively removing the reward function and assigning equal rewards to all pseudo-labels. The results from this table clearly demonstrate that eliminating the reward function leads to an increased error rate in the model, thereby underscoring the critical role of the reward function in enhancing model performance.
>
> **About lines 228-231 and how the weights are defined**: We use a uniform probability distribution to represent the least informative prediction outcome in a reinforcement learning framework, where each class is considered equally likely, reflecting maximum uncertainty. The expected Kullback-Leibler (KL) divergence then measures how much the probabilistic outputs (policy outputs) of the model deviate from this non-informative, uniform distribution. By quantifying the divergence, the expected KL-divergence effectively gauges the level of informativeness or certainty in the model's predictions. This measurement is utilized as a weight for the reward function within the reinforcement learning setup. Consequently, this weighted approach incentivizes the model to produce predictions that are more distinct or discriminatory, moving away from the uniform distribution toward more informative and class-specific predictions, thereby enhancing the model's ability to discriminate effectively among different classes.
>
>
> [1] Berthelot, David, et al. "Mixmatch: A holistic approach to semi-supervised learning." Advances in neural information processing systems (NeurIPS), 2019.

---

> > ### Comment · Reviewer_mW6a · 2024-08-11
> >
> > Thank you to the authors for their response.
> >
> > However, I remain unconvinced by the rationale for using RL. The authors mentioned that "MSE loss is less sensitive to label noise compared to CE loss," which justifies using MSE as the reward function. I believe the opposite is true, as CE has better noise resistance. Therefore, building RL training based on this reward function is not convincing.
> >
> > The explanation for the lack of significant improvement in the results in Table 3 is also unconvincing. While the test error on CIFAR-10 is low, and the differences may not be significant, there is also no noticeable improvement on CIFAR-100. This improvement might be due to model optimization and parameter selection, rather than demonstrating the significant advantage of the proposed algorithm.
> >
> > The authors might have misunderstood my question 1. I asked how the proposed method enhances the model's robustness and generalization, as claimed in the original paper. The response was about the impact of the reward function on performance.
> >
> > Therefore, while this work appears interesting, there is room for improvement. I hope the authors continue their efforts to further improve the paper. I cannot recommend accepting the current version and keep my initial scores.

---

> > > ### Author Response · Authors · 2024-08-12
> > >
> > > We thank the reviewer for the response.
> > >
> > >
> > > **About MSE loss**: Many outstanding  SSL methods [1,2,4] use MSE loss on unlabeled data, and with a similar idea MixMatch [3] applies MSE to the mixup of labeled and unlabeled data. MSE is chosen because it effectively enforces consistency, helping to stabilize training and prevent overconfidence in noisy pseudo-labels. Studies like [1] show that MSE consistently yields slightly better results than cross-entropy (CE) loss on unlabeled data. The table below highlights the effect of replacing MSE loss with CE loss in our RLGSSL in terms of test errors. These results are consistent with the studies in [1]:
> > >
> > > | Dataset         |       CIFAR-100 (4000) | CIFAR-100 (10000) |
> > > |-----------------|------------------------|-------------------|
> > > | RLGSSL          | ${36.92}_{(0.45)}$     |${29.12}_{(0.20)}$ |
> > > | R(MSE->CE)      | ${37.14}_{(0.53)}$     |${31.37}_{(0.52)}$ |
> > >
> > >
> > >
> > > **About improvement on CIFAR-100**: The improvement on CIFAR-100 dataset is not marginal. As detailed in Table 1 of our manuscript, RLGSSL outperforms the second-best method, ICT, by 3.15% and 3.12% on CIFAR-100 with 4000 and 10000 labeled samples, respectively. Furthermore, in Table 3, using a WRN backbone, RLGSSL surpasses the state-of-the-art MarginMatch by 1.24% with only 2500 labeled samples on CIFAR-100. This consistent outperformance across different datasets and experimental setups clearly demonstrates the significant advantage of our proposed algorithm, RLGSSL.
> > >
> > >
> > >
> > > **About robustness and generalization**: Our model demonstrates strong generalizability and robustness, evident in its consistent performance with the lowest average test error and low standard deviation across multiple datasets. RLGSSL not only achieves significant improvements on CIFAR-100 but also outperforms state-of-the-art methods on other datasets. This consistent success across various datasets highlights the robustness and adaptability of our model, confirming its effectiveness in varied scenarios.
> > >
> > >
> > >
> > >
> > >
> > >
> > >
> > >
> > > [1] Laine, Samuli, and Timo Aila. "Temporal Ensembling for Semi-Supervised Learning." International Conference on Learning Representations. 2017.
> > >
> > > [2] Tarvainen, Antti, and Harri Valpola. "Mean teachers are better role models: Weight-averaged consistency targets improve semi-supervised deep learning results." Advances in neural information processing systems 30 (2017).
> > >
> > > [3] Berthelot, David, et al. "Mixmatch: A holistic approach to semi-supervised learning." Advances in neural information processing systems (NeurIPS), 2019.
> > >
> > > [4] Verma, Vikas, et al. "Interpolation consistency training for semi-supervised learning." Neural Networks 145 (2022)

---

### Official Review · Reviewer_EQVV · 2024-07-04

**Soundness:** 2
**Presentation:** 3
**Contribution:** 2
**Rating:** 5
**Confidence:** 4

**Summary:**

This paper presents a method called Reinforcement Learning Guided Semi-Supervised Learning (RLGSSL), which frames SSL as a one-armed bandit problem. The method features a reward function that measures the discrepancy between the model's predictions on mixed data and pseudo-labels, guiding the learning process. Additionally, it employs a teacher-student model to enhance stability and reduce noise in pseudo-labels. The proposed joint loss function combines RL loss, supervised loss, and consistency regularization loss. Experiments validated the performance of the proposed method and the indispensability of each component.

**Strengths:**

1. This paper attempts to solve the problem of SSL from the perspective of the bandit problem, offering a fresh angle to the research.
2. The ablation study convincingly demonstrates the indispensability of each component of the proposed method.
3. The paper is written in a fluent and clear manner.

**Weaknesses:**

1. While the paper attempts to solve the SSL problem from a bandit perspective, it refers to the loss function as RL loss. The key distinction between RL and bandit problems is the presence or absence of state transitions, and the authors seem to have conflated these concepts.
2. The paper employs bandit terminology to explain parts of the methodology where it might not be necessary. Forcing SSL into a bandit framework seems somewhat unnatural, despite the novel perspective.
3. The paper appears to combine previously existing methods -- Regularization-Based Methods, Teacher-Student-Based Methods, and Pseudo-Labeling Methods. This raises questions about whether the innovation is sufficient.

**Questions:**

1. Why does Table 1 and Table 3 use different numbers of labeled samples for the same datasets, such as using 1000, 2000, 4000 for CIFAR-10 in Table 1, and 250, 1000, 4000 in Table 3? Additionally, why are the comparison methods different in these tables?

**Limitations:**

The authors have discussed limitations in their work.

---

> ### Author Rebuttal · Authors · 2024-08-06
>
> We sincerely appreciate the time and effort the reviewer has dedicated to reviewing our work.
>
> **About terminology**: The bandit problem can be viewed as a special case of reinforcement learning where there is only one transition in the trajectory. Given that the bandit problem is an older concept and less common in current literature, we formally define it as a Single-Step Markov Decision Process (SSMDP), as described in [1], to improve clarity. In this framework, we focus on maximizing the instant reward rather than the cumulative reward over multiple transitions.
>
> **About the necessity of bandit framework**: We use Reinforcement Learning (RL) and specifically one-armed bandit framework to enhance the exploration of pseudo-labels.We design the training procedure for the SSL predictor as the training of a policy function in RL. This approach enhances the predictor's performance by incorporating a novel, non-differentiable RL loss. We conducted an ablation study in Table 4 of our manuscript by dropping the RL which is the variant “$\mathcal{R}:$ w/o sg[$\theta$]". This variant demonstrates poorer results compared to our RLGSSL method showcasing the necessity of the bandit framework in our work.
>
> **About novelty**: Our approach, RLGSSL, is not merely a combination of existing SSL strategies but a novel application of reinforcement learning principles specifically designed to enhance SSL. RLGSSL introduces a specialized RL framework that uses a unique prediction assessment reward function to generate accurate and reliable pseudo-labels. This method innovatively incorporates an RL-based loss to leverage the strengths of RL, promoting superior generalization performance. Our extensive experiments, particularly the ablation studies shown in Table 5, highlight the effectiveness and innovation of our approach. The significant drop in model accuracy when the RL loss is removed ( in variant $- \text{w/o } \mathcal{L}_\text{rl}$ ) demonstrates the integral role of this component, confirming that RLGSSL is a fundamentally new method that transforms traditional SSL dynamics.
>
>
> **About the difference in tables**: The research field of standard semi-supervised learning (SSL) is extensive, with various papers employing different experimental setups. In this work, we have aimed to compare our method against a broad spectrum of standard SSL research. The primary difference between the setups in Table 1 and Table 3 lies in the backbone used. We rely on the results as reported in the related works in their respective papers. If a method is absent from any of the tables, it indicates that those authors did not provide results for that specific experimental setup.
>
> [1] M. S. Mortazavi, T. Qin, and N. Yan, “Theta-resonance: A single-step reinforcement learning method for design space exploration,” arXiv preprint arXiv:2211.02052, 2022.

---

> > ### Comment · Reviewer_EQVV · 2024-08-12
> >
> > Thank you for your response. I find the idea behind this work to be quite innovative, however, I still lean towards maintaining my original score.
> >
> > Regarding the use of the bandit framework, it seems more like a narrative tool rather than something strongly tied to the core method. Additionally, although bandit problems are a branch of RL, the primary focus and algorithms between them differ significantly. In this paper, the problem is framed using a bandit setting, yet the proposed solution is described as using an RL loss, which seems inappropriate.
> >
> > I also agree with reviewer mW6a that this paper has significant room for improvement. I hope to see a more refined version in the future.

---

> > > ### Author Response · Authors · 2024-08-13
> > >
> > > We thank the reviewer for the response.
> > >
> > >  We would like to clarify that the bandit framework is not simply a narrative tool within our study. The entire training process is built upon the bandit framework, where the policy function is optimized through iterative interactions with pseudo labels based on feedback from the reward function, rather than using a simple policy gradient. Although there are differences between the solutions for bandit problems and traditional RL problems, the primary goal of the bandit problem remains to maximize the reward function. Our carefully designed KL-divergence weighted negative reward, as discussed in Section 3.1.2, is well-suited to the bandit setting and serves as an effective solution for maximizing the instant reward received by the agent through our policy.
> > >
> > > This formulation is not merely theoretical but operational, with the bandit's reward mechanism directly influencing the learning process through the RL loss. As shown in our experiments (Tables 1 and 2 in the paper), leveraging this bandit-inspired RL loss leads to measurable improvements over state-of-the-art SSL methods across multiple datasets, indicating a concrete, beneficial impact on performance rather than a superficial narrative alignment. Furthermore, the adaptation of RL loss in this context is well-founded, as our approach dynamically adjusts to both labeled and unlabeled data, akin to how RL algorithms optimize actions based on rewards—a principle core to both general RL and bandit problems.

---

> > > > ### Comment · Reviewer_EQVV · 2024-08-14
> > > >
> > > > Thank you for your reply.
> > > >
> > > > Although I still have some concerns, I do find the idea of this paper genuinely interesting, so I’ve decided to increase my score by 1 point.
> > > >
> > > > Good luck!

---

### Official Review · Reviewer_6MiT · 2024-07-09

**Soundness:** 3
**Presentation:** 2
**Contribution:** 3
**Rating:** 5
**Confidence:** 3

**Summary:**

The authors proposes a novel Reinforcement Learning (RL) Guided semi-supervised learning (SSL) method, RLGSSL, that formulates SSL as a one-armed bandit problem and deploys an innovative RL loss based on weighted reward to guide the learning process of the prediction model adaptively. The core idea is to use RL to guide the selection of informative unlabeled samples, thereby improving the learning efficiency and effectiveness of SSL models.

**Strengths:**

The authors have introduced RLGSSL, a new approach based on Reinforcement Learning that effectively handles Semi-Supervised Learning (SSL). This method uses RL to learn effective strategies for generating pseudo labels and guiding the learning process.
The authors have devised a reward function for assessing predictions that encourages the learning of accurate and reliable pseudo-labels while maintaining a balance between the usage of labeled and unlabeled data. They have also developed a new RL loss that allows the reward from pseudo-labels to be incorporated into SSL as a non-differentiable signal in a reinforced manner, promoting better generalization performance.
Furthermore, the authors have investigated integration frameworks that combine the power of both RL loss and standard semi-supervised loss, providing a new approach that has the potential to lead to more accurate and robust SSL models.
Extensive experiments have demonstrated that this proposed method outperforms state-of-the-art SSL approaches and validates the integration of RL strengths in SSL.

**Weaknesses:**

The integration of RL introduces additional computational overhead, which may require substantial computational resources, especially for large-scale datasets.
The current formulation assumes that the labeled and unlabeled data are drawn from the same distribution, which may not hold true in real-world scenarios. This limitation could affect the generalizability of the model.

**Questions:**

1.	How can the RLGSSL method be adapted to handle scenarios where the labeled and unlabeled data come from different distributions?
2.	How can the hyperparameter tuning process be automated or simplified to make the RLGSSL method more user-friendly and less resource-intensive?

**Limitations:**

yes

---

> ### Author Rebuttal · Authors · 2024-08-06
>
> We sincerely appreciate the time and effort the reviewer has dedicated to reviewing our work.
>
> **About computational overhead**: In training deep models, backpropagation is typically the most computationally intensive step. Our method, RLGSSL, features a non-differentiable reward function and a streamlined algorithm, ensuring minimal additional computational overhead. This contrasts with many state-of-the-art methods, which often significantly extend training times due to their complexity. The table below demonstrates that for a batch size of 32 on the CIFAR10 dataset using a WRN28 backbone, the Cuda time for RLGSSL is significantly lower than that for other methods in a single training iteration. This underscores its efficiency and reduced computational cost.
>
> | Algorithm      | CPU time | Cuda time   |
> |----------------|----------|-------------|
> | MixMatch       | 5.549s   | 51.039ms    |
> | FixMatch       | 5.167s   | 287.179ms   |
> | UDA            | 5.196s   | 287.088ms   |
> | RLGSSL (ours)  | 5.186s   | 23.472ms    |
>
>
> **About distribution mismatch scenarios**: The RL component's focus on decision-making and optimization is broadly applicable across different types of learning tasks. Therefore, while domain-specific adjustments might be necessary, the foundational idea of RLGSSL could be extended and applied to different domains and tasks. Techniques such as domain-adaptive pretraining, where models are initially trained on a source domain and then fine-tuned on a target domain, or incorporating domain adversarial training, which encourages the model to learn features that are invariant across different domains, could be particularly effective. Exploring the integration of these strategies with RLGSSL to robustly address distribution shifts could be a promising direction for future research.
>
> **About hyperparameter tuning automation**: To automate and simplify hyperparameter tuning for the RLGSSL method, leveraging tools such as Bayesian optimization[1], Hyperband[2], and AutoML frameworks can be effective. These methods efficiently explore the parameter space by balancing the exploration of new configurations with the exploitation of promising ones, thereby minimizing resource expenditure. Developing more advanced and integrated hyperparameter tuning strategies for RLGSSL could be a valuable direction for future research.
>
>
>
>
> [1] Santos, Maria. "Bayesian Optimization for Hyperparameter Tuning." Journal of Bioinformatics and Artificial Intelligence 2.2 (2022).
>
> [2] Li, Lisha, et al. "Hyperband: A novel bandit-based approach to hyperparameter optimization." Journal of Machine Learning Research 18.185 (2018).

---

### Official Review · Reviewer_5CmG · 2024-07-10

**Soundness:** 2
**Presentation:** 2
**Contribution:** 2
**Rating:** 5
**Confidence:** 3

**Summary:**

One of the bottlenecks for Semi-supervised learning (SSL) is achieving high performance with limited labeled data, as the model is often complex and needs multiple loss functions. Recently RL has been increasingly used in fine-tuning complex models with non-differentiable reward functions.

Thus with these observations, the authors proposed the Reinforcement Learning Guided Semi-Supervised Learning (RLGSSL) method, which uses RL to optimize the generation of pseudo-labels in SSL.  More specifically, the pseudo-label predictor serves as the policy function and soft pseudo-labeling acts as the actions.

Technically,  the authors formulate SSL as a one-armed bandit problem with a continuous action space and deploy a novel RL loss to guide the SSL process based on a reward function specifically designed for semi-supervised data. Moreover, they further incorporate a semi-supervised teacher-student framework to augment the RL loss with a supervised loss and a prediction consistency regularization loss, aiming to enhance learning stability and efficacy. In this way, RL could provide exploration and manage non-differentiable operations.

**Strengths:**

1. The idea of leveraging RL to optimize the pseudo-labels generator in Semi-Supervised Learning (SSL) is a good catch.

2. The motivation is reasonable and in the experiment section, the author conducted broad experiments to showcase the effectiveness of the proposed method.

**Weaknesses:**

1. The comparison between the proposed method and other methods is not enough. E.g. the author could showcase the uniqueness of the proposed method in related work section,

2. The presentation of this paper is below average. E.g. there should be more figures and the writing should be more concise and logical.

3. The author did not mention the side effect of bringing RL into SSL. E.g. will the training process be more time-consuming?

**Questions:**

1. The writing should be more concise,  and the logic here is quite messy.

E.g. in "We treat SSL as a special one-armed bandit problem with a continuous action space. One-armed  bandit problem can be considered a single-step Markov Decision Process (MDP) [39]. In this problem, the agent takes a single action and receives a reward based on that action. The state of the environment is not affected by the action....". This content has been repeated a few times previously, but the reader still can not find the logic that connects these sentences.

2. The research question in this paper is not quite emphasized, it should be focused on, with deeper analysis.


3. Another thing is the authors may consider presenting the novelty more explicitly.  E.g. do more comparisons between related work and the proposed methods in the introduction and related work section.


4. More related work. This is an extension of question 3, since the idea of RL to fine-tune complex models is prevailing, readers may expect the paper to show the related work of such an idea, e.g. in the paper, the authors said "Recently, RL has been applied to fine-tune complex models that typically fail to align with users’ preferences.", then the reader would expect more related work here. And if there exists a method that tries to combine SSL and RL in any sense, then it should appear in the baseline as well.

**Limitations:**

The authors did not mention the limitations of the proposed method.


A possible limitation may fall on the training efficiency. E.g. according to Figure 1, there are multiple training loss and training objects.

---

> ### Author Rebuttal · Authors · 2024-08-06
>
> We sincerely appreciate the time and effort the reviewer has dedicated to reviewing our work.
>
> **About visualization**: We will add some figures to help visualize the results in future revisions of our paper. Nevertheless, the effectiveness of SSL is well captured in the test accuracy results reported, which is a primary and dominating evaluation norm in the literature.
>
> **About time complexity**: In training deep models, backpropagation is typically the most computationally intensive step. Our method, RLGSSL, features a non-differentiable reward function and a streamlined algorithm, ensuring minimal additional computational overhead. This contrasts with many state-of-the-art methods, which often significantly extend training times due to their complexity. The table below demonstrates that for a batch size of 32 on the CIFAR10 dataset using a WRN28 backbone, the Cuda time for RLGSSL is significantly lower than that for other methods in a single training iteration. This underscores its efficiency and reduced computational cost.
>
> | Algorithm      | CPU time | Cuda time   |
> |----------------|----------|-------------|
> | MixMatch       | 5.549s   | 51.039ms    |
> | FixMatch       | 5.167s   | 287.179ms   |
> | UDA            | 5.196s   | 287.088ms   |
> | RLGSSL (ours)  | 5.186s   | 23.472ms    |
>
>
>
>
>
> **About the research question**: In the 'Problem Setup' section, we formally and clearly define the research problem our study addresses, ensuring that readers clearly understand the framework and parameters within which our findings operate.
>
> **About more related works**: The 'Reinforcement Learning' subsection of  Related works in the manuscript provides a detailed overview of various studies that employ reinforcement learning (RL) techniques to tackle diverse challenges. We will expand the related work section to include additional studies that apply reinforcement learning (RL) to fine-tune complex models. It is important to note that to the best of our knowledge, our work is the first to use RL to guide semi-supervised learning (SSL), which represents the novelty of our approach.
>
> **About limitation**: The limitation section can be found in Appendix E of the manuscript.
>
> **About loss terms**: Our RLGSSL method adds just one extra loss term—the RL loss—to the standard framework of semi-supervised learning (SSL), which typically includes supervised and consistency losses.  The non-differentiable nature of the reward function within this reinforcement learning (RL) loss ensures that it does not significantly increase computational overhead during training (as shown above). Furthermore, the only trainable entity is the student model; the teacher model's parameters are updated using an Exponential Moving Average (EMA), which simplifies the training process. This streamlined approach addresses concerns about training efficiency,

---

> > ### Comment · Reviewer_5CmG · 2024-08-12
> >
> > Thanks for the response. The Rebuttal mitigates my concern about training efficiency. So I will keep my positive opinion of this paper.

---

### Decision · Program_Chairs · 2024-09-25

**Decision:**

Accept (poster)

**Comment:**

The paper introduces a novel method to combine reinforcement learning (RL) with semi-supervised learning (SSL) to improve the generation of pseudo-labels. Particularly, the key insight is to first mix-up labeled and unlabled covariates, and also their label and pseudo-labels, respectively. Subsequently, the label predicted for the mixed-up covariate is made to match its mixed up label obtained from the previous step. This provides a new consistency loss for the label predictor. This loss is subsequently weighted by a KL term that encourages the labels to be distinctive, thereby preventing oversmoothing of the labels. This results in a complex objective that is treated as a non-differentiable reward to be optimized for. While reviewers raised some concerns regarding higher compute requirement when using  RL methods, authors provided in rebuttal reports of favorable compute time. Some more questions were raised regarding the choice of consistency loss (MSE vs cross entropy), and the authors provided additional experimental results in the rebuttal to justify their design choice.

Overall, the insights in the paper are interesting, and the gains in the performance are significant.

------
Some additional experiments to consider: I wonder if the method can provide even better result if the consistency loss is created for not only unlabeled-labeled pair, but also for labeled pairs of data?